# Magma Chambers and Meteoric Fluid Flows Beneath the Atka Volcanic Complex (Aleutian Islands) Inferred from Local Earthquake Tomography

**Ivan Koulakov [1,2,3,*], Ekaterina Boychenko [4] and Sergey Z. Smirnov [5]**

[1]   Trofimuk Institute of Petroleum Geology and Geophysics SB RAS, Prospekt Koptyuga, 3, Novosibirsk 630090, Russia

[2]   Department of Geology and Geophysics, Novosibirsk State University, Pirogova 2, Novosibirsk 630090, Russia

[3]   Institute of Volcanology and Seismology FEB RAS, Piip Boulevard, 9, Petropavlovsk-Kamchatsky 693006, Russia

[4]   Department of Physics and Technology, Novosibirsk State Technical University, Karl Marks Avenue 20, Novosibirsk 630073, Russia; boych.ek@yandex.ru

[5]   Sobolev Institute of Geology and Mineralogy SB RAS, Prospekt Koptyuga, 3, Novosibirsk 630090, Russia; ssmr@igm.nsc.ru

**\***   Correspondence: KoulakovIY@ipgg.sbras.ru

**Abstract:** Atka is a subduction-related volcanic island located in the central part of Aleutian Arc. The northeastern part of this island forms the Atka Volcanic Complex (AVC), which is built as a relict shield volcano of a circular shape overlain by several active and extinct volcanic vents of different ages. During the past few decades, two active volcanoes within AVC,—Korovin and Kliuchef—demonstrated mostly phreatic eruptions and intensive fumarolic activity. We have created the first tomographic model of the crust beneath AVC with the use of data of eight permanent stations of the Alaskan Volcanological Observatory operated in the time period from 2004 to 2017 that included arrival times of the P and S waves from local seismicity. Based on a series of checkerboard tests, we have demonstrated fair vertical and horizontal resolution of the model down to ~6 km depth. Beneath the Korovin and Kliuchef volcanoes, we have revealed two isolated anomalies of high Vp/Vs with values exceeding 2, which represent separate magma chambers that are responsible for magmatic eruptions of these two volcanoes. In shallow layers down to 2–3 km deep, we observe an alternation of zones with low and high values of the Vp/Vs ratio, which are likely associated with the circulation of meteoric fluids in the uppermost crust. Moderately high Vp/Vs anomalies indicate zones of meteoric water penetration down to the ground. On the other hand, the very low values of Vp/Vs reaching 1.5 depict the areas where meteoric water reached the hot magma reservoir and transformed into steam. On the surface, these zones coincide with the distributions of fumaroles. The outflow of these steam currents from active vents of Korovin and Kliuchef led to episodic phreatic eruptions, sometimes synchronous.

**Keywords:** Aleutian Arc; Atka Island; Korovin Volcano; seismic tomography; local seismicity; magma chamber; meteoric fluids.

## 1. Introduction

Aleutian Islands (Figure 1a) is an area where high volcanic and seismic activity may potentially pose serious problems for the entire North Pacific region [1,2]. In particular, many volcanoes in Aleutians have traces of violent explosive eruptions in the recent geological past [3]. Each of these events may become a serious problem for local people and air transportation [4]. To monitor volcanic

and seismic activity, most of active volcanoes in Aleutians are equipped with networks of telemetric seismic stations that transmit the continuous data in real time to the AVO office [5]. These stations, which operate for decades, give valuable information that can be used to investigate the inner structures of the volcanoes. The estimated seismic models for some of the volcanoes for the Alaskan-Aleutian region help better understand the mechanisms of plumbing systems [6–9].

In this study, we present the results of seismic study for the Atka Volcanic Complex (AVC) belonging to the Atka Island. It is located in the Adreanoff segment of the Aleutian arc, together with five other volcanic islands of very different composition and eruption styles [10]: Kanaga, Adak, Great Sitkin and Seguam, of which Atka Island is the largest. This chain of volcanoes is associated with the ongoing subduction of the Pacific Plate, which is slightly oblique in this segment, and has a rate of ~87 mm/yr. Myers et al. [11] noticed that the Aleutian volcanoes stay fixed in one location for a long time of around 5 Ma. Therefore, beneath most of volcanic centers, steady magma pathways were established, which determined a long history of their development.

Morphologically, the Atka Island consists of two geographically distinct parts (Figure 1b). The elongated southwestern part is 85 km long and 10 km in width. It is mostly composed of igneous and volcanoclastic material and metamorphic rocks without any manifestations of Holocene volcanism [12]. The northeastern part of the island is a distinct circular area covering 360 km$^2$, which is connected with the rest of the island by a narrow isthmus separating Korovin and Nazan Bays (Figure 1c). This part is composed of volcanic rocks, with the total erupted volume reaching 200 km$^3$. The age of the oldest rocks in Atka is approximately 6.6 Ma, based on the K-Ar dating method [13]. The circular shape of this part of the island was formed 1–2 Ma BP, by basaltic and basalt-andesitic flows originated from a single eruptive center [13,14]. This resulted in creating a large shield volcano with the basal diameter of ~20 km and the altitude probably reaching 2200 m [15]. Then, the basaltic activity evolved to more silicic type occurring from several distinct vents. In particular, a massive eruption of Big Pink dacites (BPD in Figure 1c) occurred in the interval of 0.3–0.5 Ma, which led to forming a large caldera in the central part of the shield volcano, with the diameter of ~5 km. Simultaneously with this event, the entire Atka shield volcano was cut by a series of dikes [15].

The post-caldera activity of AVC occurred in four major volcanic cones: Korovin, Konia, Kliuchef and Sarichef. Presently, Korovin is the most active and highest stratovolcano, reaching the altitude of 1530 m [16]. A lack of traces of glacial erosion indicate that the age of this volcano does not exceed 100,000 years [15]. Korovin has two distinct summits 0.6 km apart. The northwestern summit has a small crater. The crater of the southeastern summit is much larger: 1 km wide and several hundred meters deep. Most of time, in the bottom of southeastern crater, there is a steaming lake, which sometimes disappears, usually prior to the phreatic eruptions of Korovin [15].

Another active volcano of the AVC, Mount Kliuchef (1460 m) is located within the caldera and is composed of basalt-to-dacite lava flows and pyroclasts [15,16]. The southern and western flanks of Kliuchef are presently marked by active thermal fields and hot springs (yellow fields in Figure 1c) [17]. Konia is another predominantly dacitic stratovolcano between Korovin and Kliuchef having a smaller size. The recent activity of Konia is evidenced by a fresh cinder cone on the western flank of this volcano. The fourth major volcano of AVC is Sarichev located at the eastern side of Atka. No robust evidence for any recent eruption activity of Sarichef is recorded. At the perimeter of the AVC, there are traces of several other vents (indicated by M1 to M6 in Figure 1c) strongly dissected by glacial processes, which lay over the shield-volcano lavas and may be remnants of pre-caldera stratovolcano edifice.

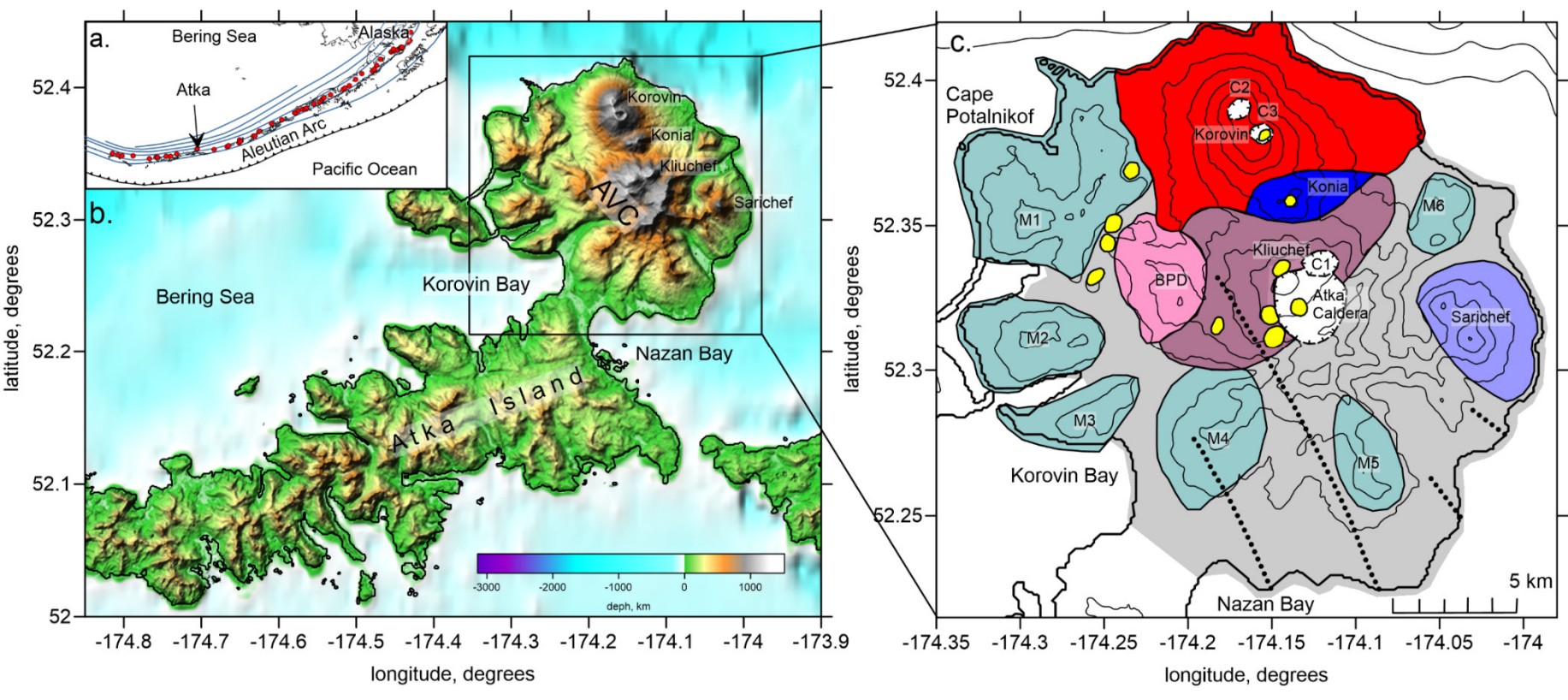

**Figure 1.** General framework of the study area. **a**. Aleutian arc and Holocene volcanoes (red dots). The location of the subducting slab is indicated by grey contour lines at every 50 km depth. **b**. Topography of the Atka Island. AVC is Atka Volcanic Complex. **c**. Major structural elements within the AVC. White areas depict volcanic craters and caldera. Yellow areas indicate geothermal fields. Light grey background indicates the location of the Pra-Atka shield volcano. Light blue areas indicated by M1 to M6 highlight areas of extinct vents. BPD is a place of the Big Pink Dacite flow. Dotted lines are the dike systems.

According to Marsh [18], more than 90% of rocks in the AVC are basalts with a silica content of around 50 wt%, which mostly corresponds to the earlier stages of the shield volcano development. In the later stages, the silicic content gradually increased up to 66 wt%, expanding the composition variability to andesites and dacites. Geochemical analyses by [15] indicate that the lavas erupted from Korovin and Kliuchef volcanoes differ in their trace-element evolution patterns. This may suggest that these stratovolcanoes are fed from different non-communicating crustal magma reservoirs [15]. On the other hand, there are records of synchronous phreatic eruptions of the Korovin and Kliuchef volcanoes [3], indicating some interconnection between the magma and hydrothermal sources. Therefore, the question about the geometry of the feeding system beneath AVC remains open.

Starting from 1829, there are episodic observations of the eruption activity of Korovin. However, the actual number of eruptions can be much larger than the recorded number, due to the inaccessibility of this area and the endemic poor visibility of the volcano. Strong steam emissions in Korovin were recorded on 23 May 1986 soon after a 7.7 magnitude earthquake, which stroke at approximately 100 km distance [19]. On 18 March 1987, the ash plumes were simultaneously ejected from Korovin and several Kliuchef vents [3], which might indicate interconnection of their plumbing systems. In the period between 1998 and 2007, the Korovin volcano manifested an activity increase, which resulted in moderate short-lived phreatic eruptions ejecting ash plumes up to several kilometers. From 2004, the installed AVO network of telemetric seismic stations [5] started regular recording seismicity within AVC. In correlation with phreatic eruptions occurred in that time, this network detected several periods of elevated seismic activity, with some local earthquakes with magnitudes reaching up to 3.5 and tremor occurrence [20,21]. In 2005–2007, consistently with this period of activity, the analysis of the InSAR images has detected more than 8 cm of uplift in Atka [22]. Zhan et al. [23] created a numerical model simulating this uplift, in which the source of deformation took place beneath the center of AVC at the depth of ~5 km below the surface.

Important information about the ongoing processes in the magmatic system beneath the area of AVC was derived from the locations of seismicity recorded by eight telemetric stations starting from 2004 [5]. However, the distributions of the earthquakes beneath AVC were sparse and strongly scattered, and they alone did not allow identifying the structure of the magma reservoirs. No other geophysical studies, which could provide images of the plumbing system beneath AVC, were previously performed. In this study, we present first images of the upper crust structures beneath the AVC based on local earthquake seismic tomography. The robustness of the results is supported by a number of synthetic tests showing fair resolution of the results.

## 2. Data and Algorithm

In this study, we used the data of catalogues with the information on local seismicity and arrival phase data for the years from 2004 to 2017 provided by the Alaskan Volcano Observatory (AVO). For the Atka Island, we used, in total, eight permanent stations: seven stations of AVO installed in July 2004 [5] and one station of the Alaska Earthquake Center installed in 2002 on another side of the isthmus in the town area. In total, the catalogue included the information on 3398 events with 14,366 P and 11,496 S wave arrival times (on average 7.6 picks per event). When selecting the events for tomography, we used several criteria: (1) the distance from an event to the nearest station should not be larger than 20 km; (2) the total number of the P and S wave picks per event should be equal to or larger than 4; (3) the time residual after locating the events in the starting 1D velocity model should not be larger than 0.5 s. After applying these criteria, the number of data slightly reduced to 2946 events, 12,036 P and 10,427 S picks. The locations of the events after the running the full tomography procedure are shown in map view, and in four vertical sections in Figure 2.

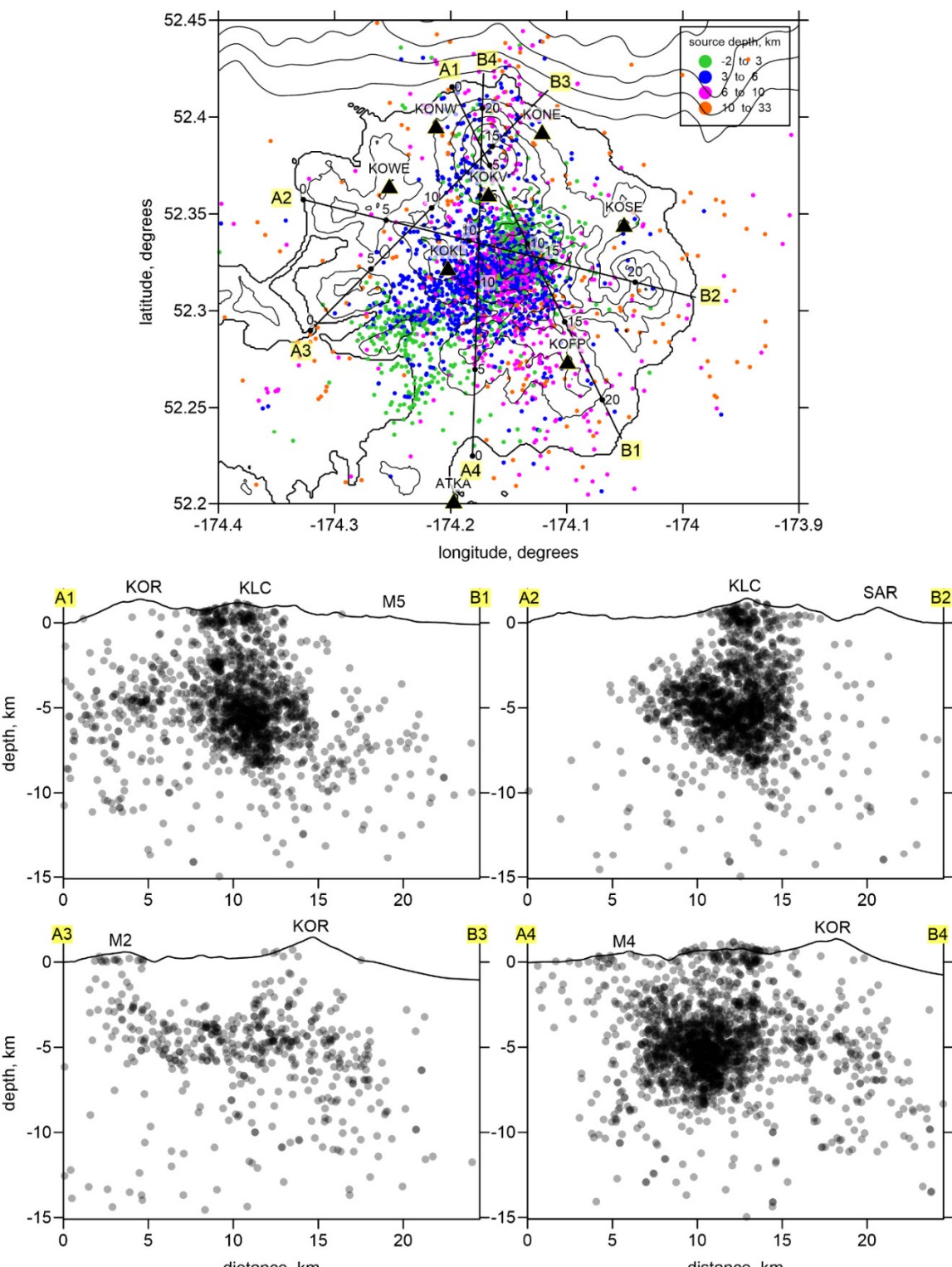

**Figure 2.** The final locations of the events after five iterations of tomographic inversion in map view and vertical sections. In the map, the black triangles depict seismic stations. Colored dots indicate the seismicity distribution according to the depth. Topography is indicated with contour lines at every 250 m. In vertical sections, the events at distances of less than 3 km from the profile are shown. Volcanoes: KOR-Korovin, KLC-Kliuchef; SAR-Sarichef, M1 to M6 are the same massifs as indicated in Figure 1c.

There might be some concern about the low number of stations and the low ratio of picks per event in our case, which could potentially prevent the obtaining of high quality tomography results. Previously, there were several studies with similar configurations of stations, such as 8 stations on

the Avacha volcano [24], 8 stations on Nevado del Huila [25] and even 4 stations on Udina [26], which successfully revealed useful information on the underground structures based on seismic tomography. In all these cases, careful testing was performed that demonstrated fair robustness of the resulting models. In this study, we do a similar job of performing synthetic modeling, which will be presented later and will show the resolution limitations for our models.

In this study, we use the same algorithm LOTOS [27] and same workflow as implemented in the studies with limited numbers of stations mentioned above, as well as in many other cases. This software has been described in a number of previous publications; therefore, we only briefly list the major steps and features of the algorithm here.

The calculations start with preliminary locations of the sources, which uses the grid-search method enabling a stable determination of source coordinates even if no a priori information is available. We use a successive refining of the grid for searching the best source location using three grids with spacing of 4 × 4 × 4 km, 1 × 1 × 1 km and 0.1 × 0.1 × 0.1 km. For each point of the grid, we calculated an objective function representing the probability of source location in the current point (see details in [27]). When reaching an extremum value of the objective function in a coarser grid, we perform the further search in a finer grid. At this stage, to calculate the travel times, we use straight ray paths, which makes it possible to speed up voluminous grid search calculations.

In the next step, we repeat the stage of source locations, but with the use of the 3D ray tracer based on the bending method, the concept of which was proposed by Um and Thurber [28]. The ray paths are calculated between sources and stations with actual elevations. In the first iteration, this step is performed for the starting 1D model; in the next iterations, the sources are relocated in the updated 3D velocity model. Note that, in all stages, the realistic relief is used to limit the source distributions so that they can be placed below or above the sea level, but cannot be above the topography surface.

The three-dimensional distributions of the P and S wave velocity anomalies are parameterized by a set of nodes distributed in the study area according to the distributions of the ray paths. In map view, the nodes are placed in the regular grid with spacing of 1 × 1 km. However, in areas without data, the nodes are missing. In the vertical direction, the distance between nodes inversely depend on the ray density, but it cannot be smaller than a predefined value, 0.5 km in our case. These grid parameters appear to be optimal in our case: for the larger grid spacing, it would be compatible with the size of the resolved anomalies, and the model would be grid-dependent. Between nodes, the velocity is calculated using the bi-linear interpolation. Note that the predefined grid spacing is much smaller than the expected resolution of the model; every resolved anomaly is thought to be based on several nodes. In this sense, varying the parameters of the grid does not affect the resulting model. At the same time, to further decrease the effect of the grid geometry to the results, we performed a series of inversions in grids with different basic orientations (0°, 22°, 45° and 66°) and then created an average model based on a regularly spaced grid.

The inversion is performed simultaneously for the P and S wave velocity anomalies (dVp and dVs) and source corrections (three parameters for the coordinates and one for the origin time). Normally, the LOTOS code provides a possibility to include the station corrections, however, we do not use this option in our case. The inversion of the large sparse matrix is performed using the LSQR algorithm [29,30]. To avoid ill-posed problem, we add smoothing and amplitude constraints that enable damping in the resulting velocity inversion. The optimal values of the regularization parameters are determined based on synthetic modeling. After the inversion step, we update the 3D velocity model, which is used in the next iteration for the relocation of the sources. Then, the calculation of the new matrix and the inversion is repeated again. Five iterations were used in our case, which is a compromise between the calculation time and accuracy of the solution.

The 1D starting model for the inversion was defined iteratively after several runs of the full tomography procedure with the use of different reference velocities. After each run, we updated the starting model according to the average velocity values at several depth levels. As a result, in the final model, we get a balanced model with approximately equal amount of the "blue" and "red" anomalies in the horizontal sections. For the P-wave velocity model, we set 4.53 km/s at 1 km depth, 4.73 km/s

at 4 km, 4.88 km/s at 7 km, and 5.11 km/s at 10 km depth. Between these levels, the velocity values were linearly interpolated. To define the S-wave velocity in the starting model, we use a constant value of the Vp/Vs ratio (1.75 in our case) to avoid a predefined layering of this parameter. This value was set to achieve an appropriate balance of the positive and negative anomalies in the S-wave velocity model.

## 3. Tomography Results

Before presenting the tomography model calculated using the field data, we show the results of synthetic tests that are especially important in our case taking into account the low number of the stations. Furthermore, the synthetic tests are used to determine optimal values of controlling parameters for calculations when using the experimental data. Here, we separately investigate the horizontal and vertical resolution using checkerboard models defined either in map view or along the same profile as used for presenting the main model. In all cases, the synthetic data are calculated for the same source-receiver pairs as used for calculation of the experimental data model. The rays are traced through the three-dimensional synthetic model using the bending algorithm of ray tracing. The synthetic data are then perturbed by random noise. The mean value of noise depends on the size of the network and quality of the data. In our case, we defined the noise mean deviation of 0.05 s and 0.1 s for the P and S wave data, respectively. This noise level provides similar values of variance reduction in the cases of experimental and synthetic data inversions. After calculating the travel times, we "forget" the coordinates and origin times of the sources, and start the recovery procedure with grid-search location of the source coordinates using the starting 1D model, similarly to what we did to obtain the main model. Then, we reproduce the same inversion workflow and use the same controlling parameters as in the case of the experimental data processing.

In Figure 3, we present the checkerboard test for assessing the horizontal resolution. In the presented case, the alternated squared anomalies of ±7% magnitude with the size of 4 × 4 km are defined. With depth, these anomalies remain unchanged. We can see that the Vp and Vs anomalies are recovered with correct shapes and amplitudes at depths of 0 and 3 km. At the depth of 7 km, general configuration of anomalies is revealed correctly; however, they are strongly smoothed and have approximately twice lower amplitudes compared to the original model. The poorer resolution can be explained by a significant lower amount of seismicity below this level. Nevertheless, on a qualitative level, the anomalies at this depth can be used for interpretation. Another important conclusion of this test is that the Vp/Vs ratio, which is simply calculated from division of the recovered absolute values of Vp over Vs, is robustly resolved in this model.

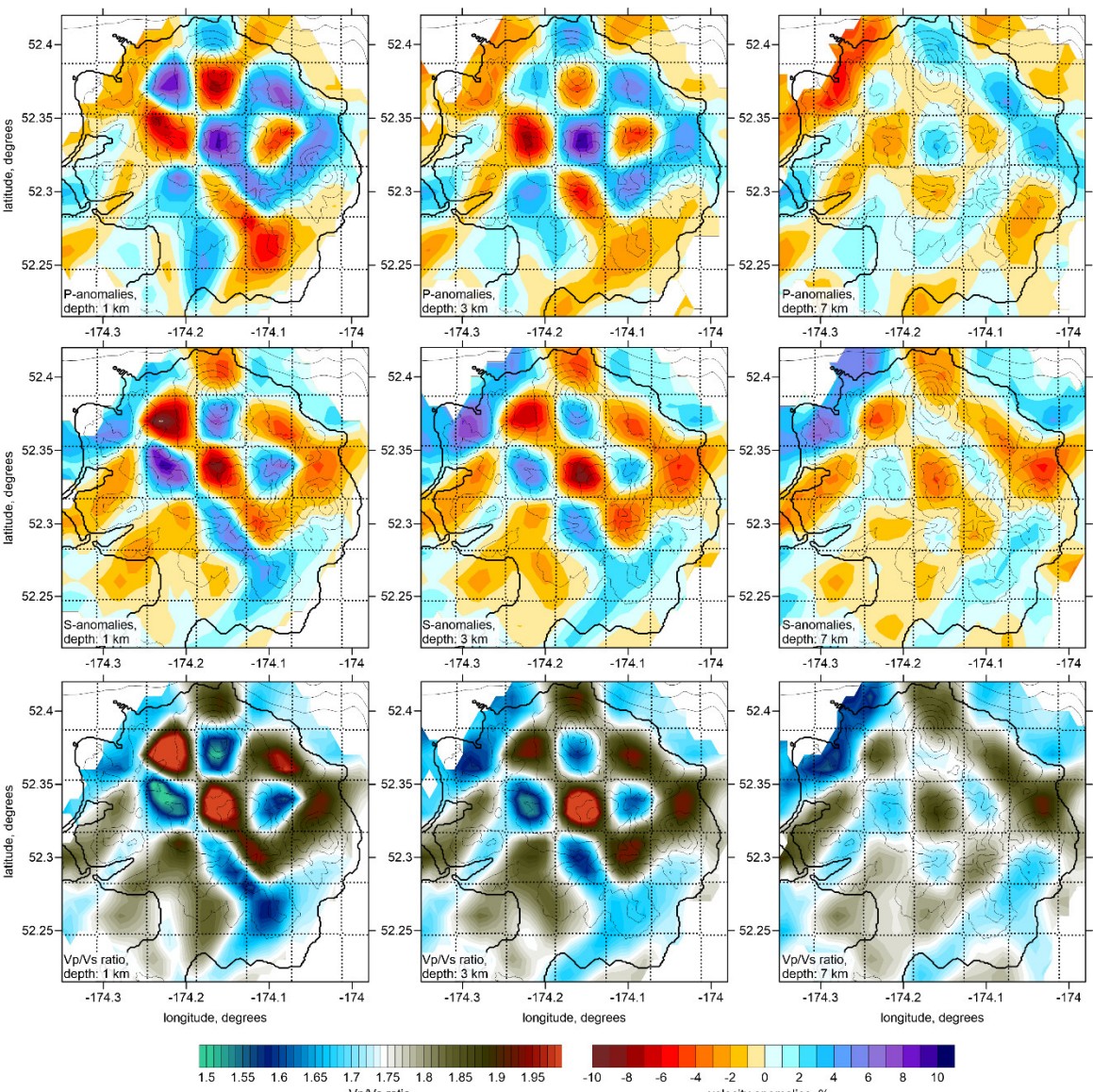

**Figure 3.** Checkerboard test for checking the horizontal resolution. The recovered distributions of the Vp and Vs anomalies and Vp/Vs ratio are shown in three horizontal sections. The locations of the initial anomalies are marked with dotted lines. The topography is shown with contour lines with the interval of 250 m.

To assess the vertical resolution, we performed a series of tests with anomalies defined along vertical sections, the same as were used for presenting the main results (Figure 4). For each section, we create a separate checkerboard model with squared anomalies of 4 × 4 km, which change the sign at the depths of 2 km, 6 km, 10 km etc. Across the section, the anomalies have the width of 5 km and remain unchanged. We see that in all sections, the dVp, dVs and the Vp/Vs ratio are recovered correctly in the upper two layers down to 6 km depth. Below this level, the obtained anomalies are strongly smeared, although, in most cases, they appear to be at right locations.

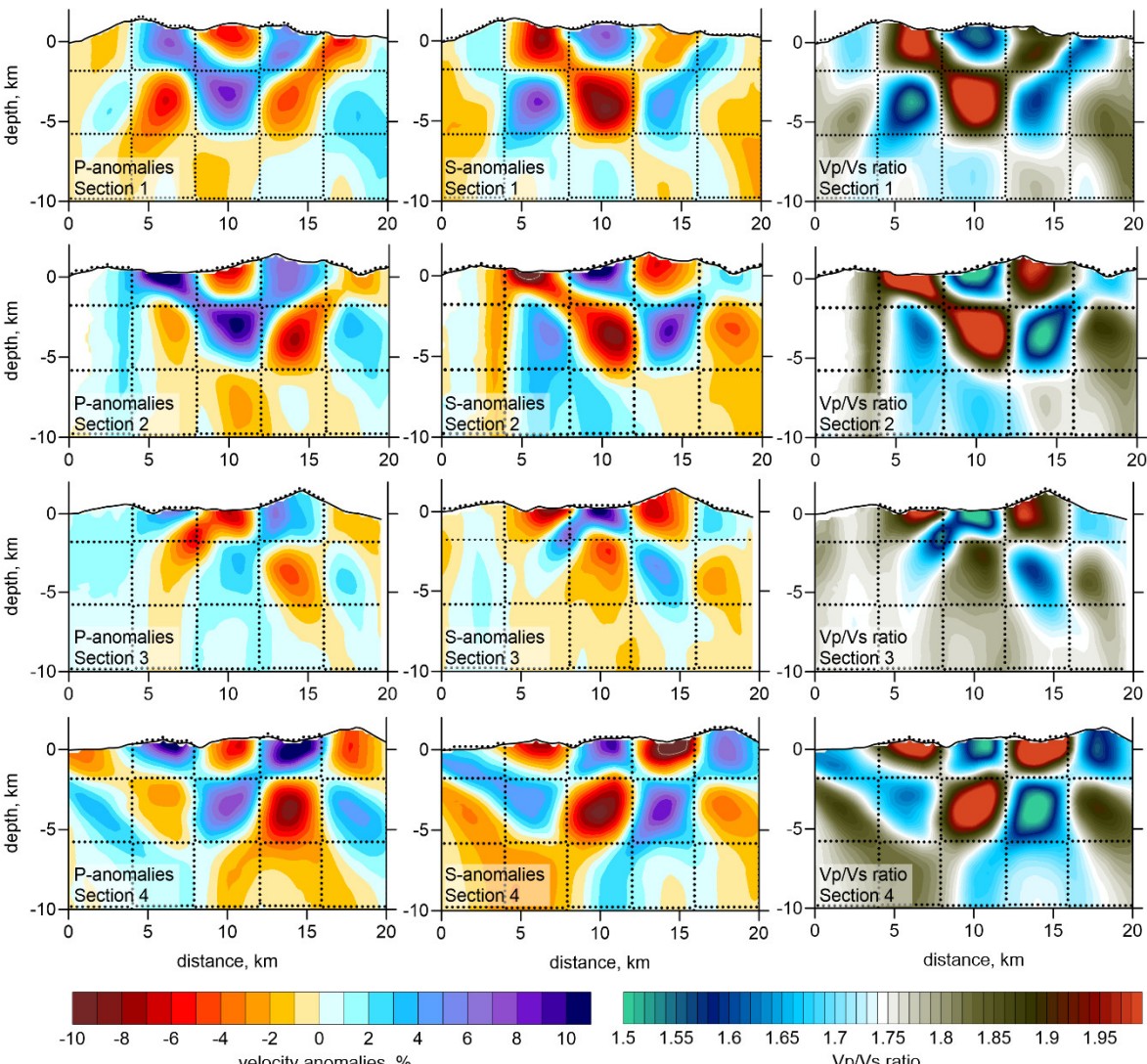

**Figure 4.** Checkerboard tests for checking the vertical resolution. Four different models are defined along four sections, same as shown in Figure 2. The locations of the initial synthetic anomalies are indicated with dotted lines.

The main results of this study, which were derived from the inversion of the experimental data, are presented as dVp, dVs and Vp/Vs ratio in three depth levels (Figure 5) and four horizontal sections passing through the main volcanic structures of AVC (Figure 6). The distributions of the earthquakes relocated in the obtained three-dimensional model are presented in map view and in the same vertical sections in Figure 2. The interpretation of these results will be given in the next section.

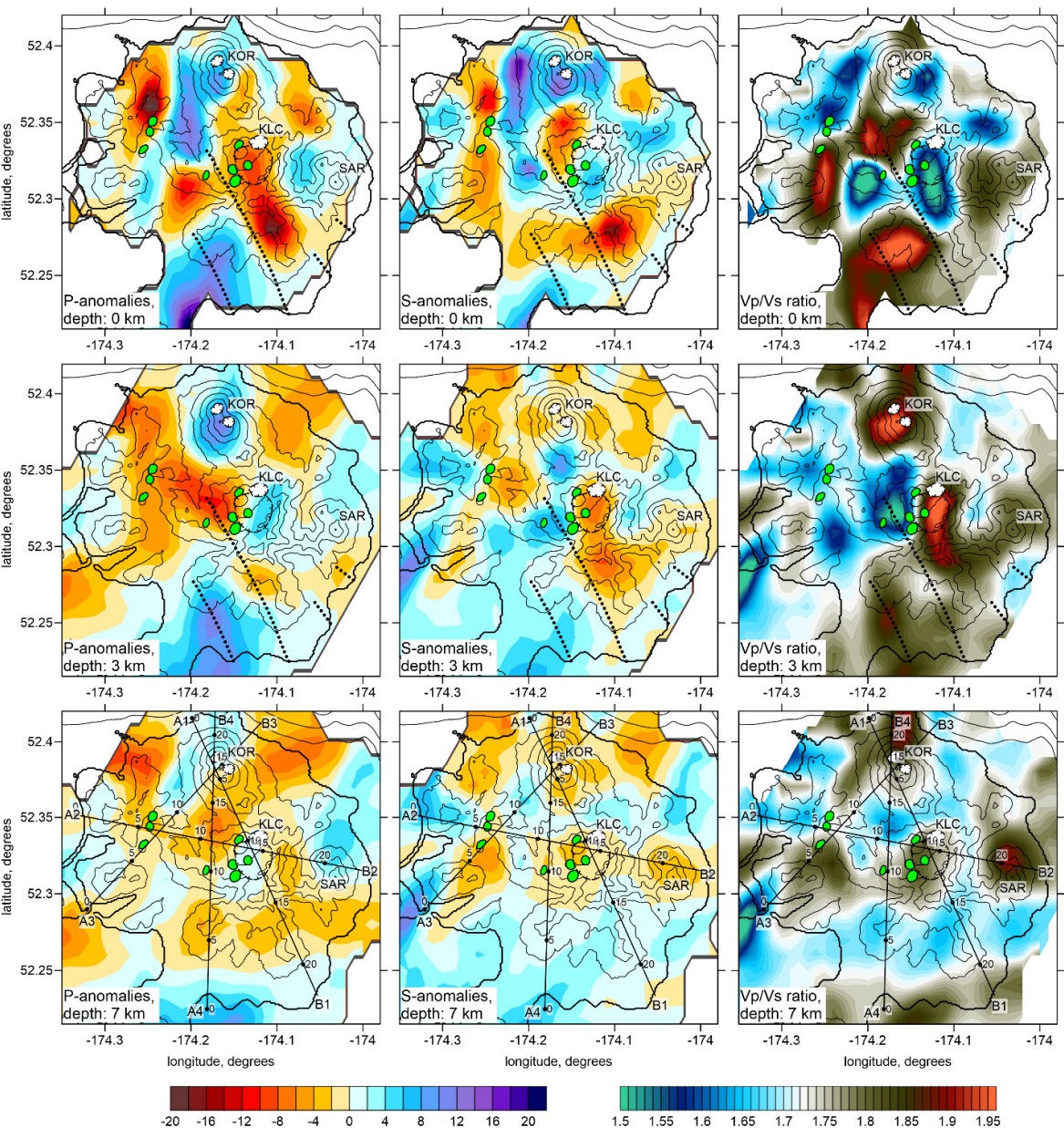

**Figure 5.** The resulting distributions of the dVp, dVs and Vp/Vs ratio derived from the inversion of experimental data in three horizontal sections. Green areas depict geothermal fields; white areas are the main craters of the active volcanoes. The topography is shown with contour lines, with the interval of 250 m. Volcano indications: KOR-Korovin, KLC-Kliuchef and SAR-Sarichef. Dotted lines mark the dike systems. Lines indicate the locations of four vertical sections shown in Figure 6.

## 4. Discussion

The derived seismic anomalies and especially the distribution of the Vp/Vs ratio are the important indicators for the presence of fluids and melts, which are the key factors controlling the state of active magma systems [9,31,32]. The obtained results demonstrate clear correspondence of the seismic anomalies with the main volcanic structures in AVC.

At shallow depths (section 0 km in Figure 5), the Korovin volcano is associated with high dVp and dVs. This is quite a common feature for volcanoes with a predominantly mafic composition, representing the existence of a rigid body within the stratovolcano composed of igneous rocks. Similar high-velocity structures were previously observed in many volcanoes of the world, such as Mt. Vesuvius in Italy [33], Redoubt volcano in Alaska [8], Popocatépetl volcano in Mexico [34] and others. At the depth of 3 km, the P-wave velocity remains high beneath Korovin, whereas the S-wave

velocity anomaly turns out to be negative, which results in a prominent high-Vp/Vs ratio reaching 2. Such a combination of seismic parameters is a clear indicator for the presence of a liquid phase, such as dissolved volatiles or melts [9,31,32,35]. The same pattern of high Vp, low Vs and high Vp/Vs ratio is observed at the depth of 7 km; however, the high-Vp/Vs anomaly is slightly shifted with respect to the volcano's summit, and is located beneath its northern flank.

In the vertical sections 1, 3 and 4 (Figure 6) the anomaly of very high Vp/Vs beneath Korovin appears as an isometrical body at depths between 2 and 5 km below surface. We propose that this anomaly represents the active magma chamber, which is responsible for the present eruptions of this volcano. The upper limit of this anomaly appears to be nearly parallel to the topography line. A very similar feature was identified beneath another volcano of similar composition and degassing style, Gorely in Kamchatka [32], as well as in some other volcanoes [31]. In those studies, it was proposed that the upper boundary of the chamber related Vp/Vs anomaly represents the limit where the dissolved in magma fluids are transformed to gases, due to crystallization of anhydrous minerals and/or decompression. Alternatively, it may represent the meteoric waters percolating down and heated, due to interaction with deeper two-phase hydrothermal system (steam-heated waters) [36] or convective heat transfer from magmatic source. As there are evidences that magmas in AVC are water unsaturated [15,37], the second scenario involving the meteoric water is preferable for describing the hydrothermal activity and frequent phreatic eruptions.

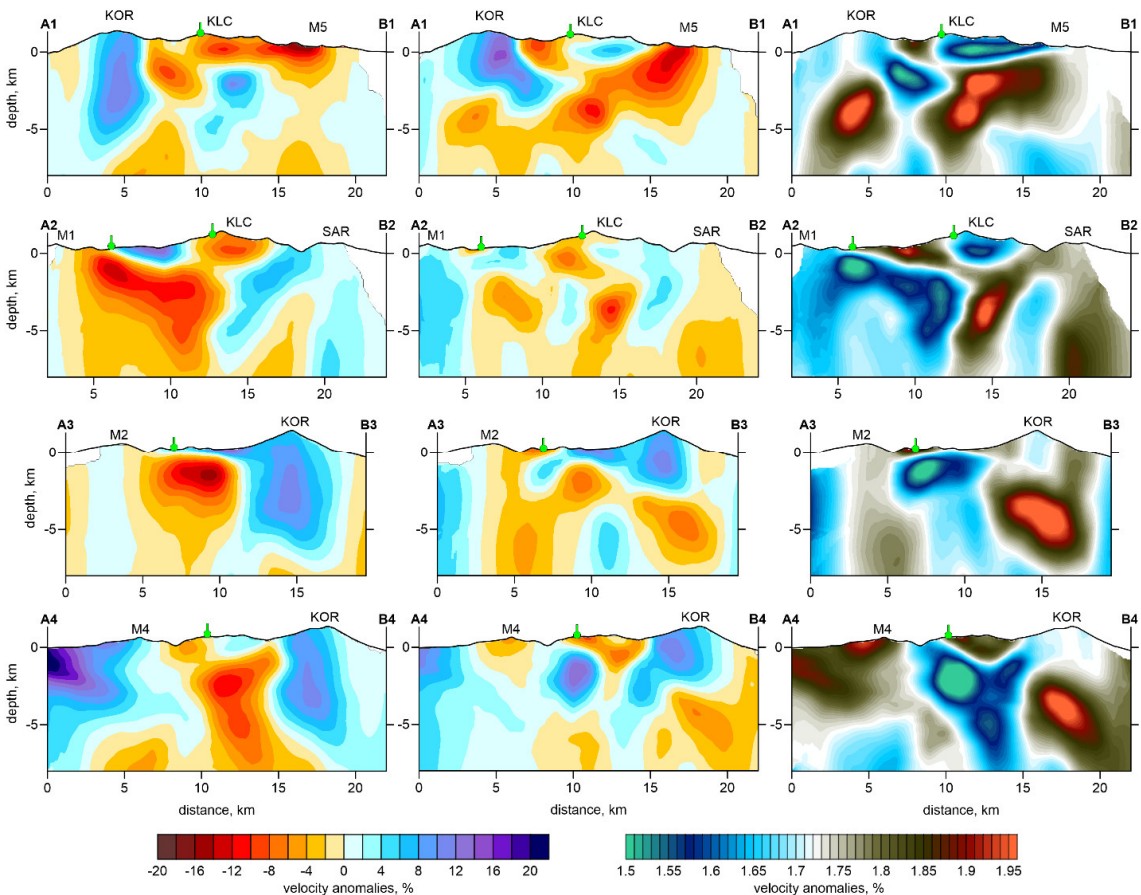

**Figure 6.** The resulting distributions of the dVp, dVs and Vp/Vs ratio derived from the inversion of experimental data in four vertical sections. Green symbols indicate the locations of geothermal fields. Volcanoes: KOR-Korovin, KLC-Kliuchef; SAR-Sarichef, M1 to M6 are the same massifs as indicated in Figure 1c.

Beneath the Kliuchef volcano and Atka Caldera, at the level of 0 km (Figure 5), we observe a prominent anomaly of low dVp and high dVs, which gives very low Vp/Vs ratio at around 1.5. This is a typical coexistence of seismic parameters for gas-saturated rocks [31,32,38], as they behave as a sponge having very low bulk elastic parameters [39]. This hypothesis is supported by the existence

of several hydrothermal fields in the caldera area [17], which match this anomaly of low Vp/Vs ratio. At greater depth, the relationships between the P and S wave parameters turn out to be opposite beneath the Kliuchef and the Atka Caldera, at 3 km depth, we observe a prominent high Vp, low Vs and very high Vp/Vs ratio, which is similar to the structure beneath Korovin at the same depth. At 7 km depth, the Vp anomaly becomes less prominent, but still positive, and the dVs is negative that gives elevated Vp/Vs ratio. At the same time, it should be remembered that the synthetic tests demonstrated a significant smearing of anomalies at this depth, meaning that the magnitudes of anomalies in our results at this level could be underestimated.

In the vertical sections 1 and 2 (Figure 6), the anomaly of high Vp/Vs ratio beneath Kliuchef and Atka Caldera appears to more complicated in shape than that beneath Korovin. In section 1, the upper limit of this anomaly is at ~1.5 km depth below the caldera and it gradually gets shallower to the right part of the section in the area of the Massif M5. We also see, in Section 1, that the anomaly of high Vp/Vs ratio beneath Kliuchef appears to be separated in two parts, below and above the limit at the depth of ~4 km. In section 2, this anomaly of high Vp/Vs ratio beneath the caldera looks slightly inclined in the deeper part, whereas in the shallow part it appears to be split in two branches surrounding the body of the Kliuchef volcano.

It is interesting to compare the location of this anomaly of high Vp/Vs ratio beneath the Atka Caldera with the observations of ground deformations, which demonstrated strong uplift up to 8 cm in 2005–2007 in the central part of the AVC [22]. The numerical model developed by Zhan et al. [23] has provided an estimate for the magma source location. Laterally, it corresponds fairly well to the location of the magma chamber beneath the Atka Caldera (see the section at 3 km depth in Figure 5). However, vertically, they provided the depth of the magma chamber at 5 km below surface. Note that, in our result, the Atka magma chamber consists of two distinct parts: one from 2 to 4 km, and another one from 4 to 6 km. It is possible that the deformation recorded in 2005–2007 was caused by the intrusion of a fresh magma portion to the deeper part of the reservoir that caused the observed uplift.

This two-level structure of Vp/Vs anomaly is consistent with the model of [15], which suggests that later dacites of Kliuchef volcano could be a result of partial melting of older Atka rocks. The lower part of the twin anomaly could be interpreted as an intrusion of hot and more basic magma, while the upper one could be the area of partial melting and mixing between siliceous partial melts and more basic melts penetrating from lower magma reservoir.

Aleutian volcanism is dated back from 38 Ma, and is divided into three high activity pulses: 38–29 Ma, 16–11 Ma and, more recently, 6–0 Ma [40]. The AVC belong to the youngest pulse that started at 6 Ma and, according to [11,41], occurs very steadily, without a change of locations of the main volcanic centers for very long time, possibly for several millions of years. The large shield volcano determining the shape of the circular northeastern part of the Atka Island was formed 1–2 million years ago. Although, during the evolution of this complex, volcanic manifestations took place in different points of this circular area, the major activity always occurred beneath the central point, coinciding with the present location of the Kliuchef volcano and Atka Caldera. Therefore, we propose that the prominent seismic anomaly indicated in Figure 7 as Atka magma chamber, is the top of a well-established and very long lived conduit, which possibly controlled the volcanic activity of this complex for at least a million of years.

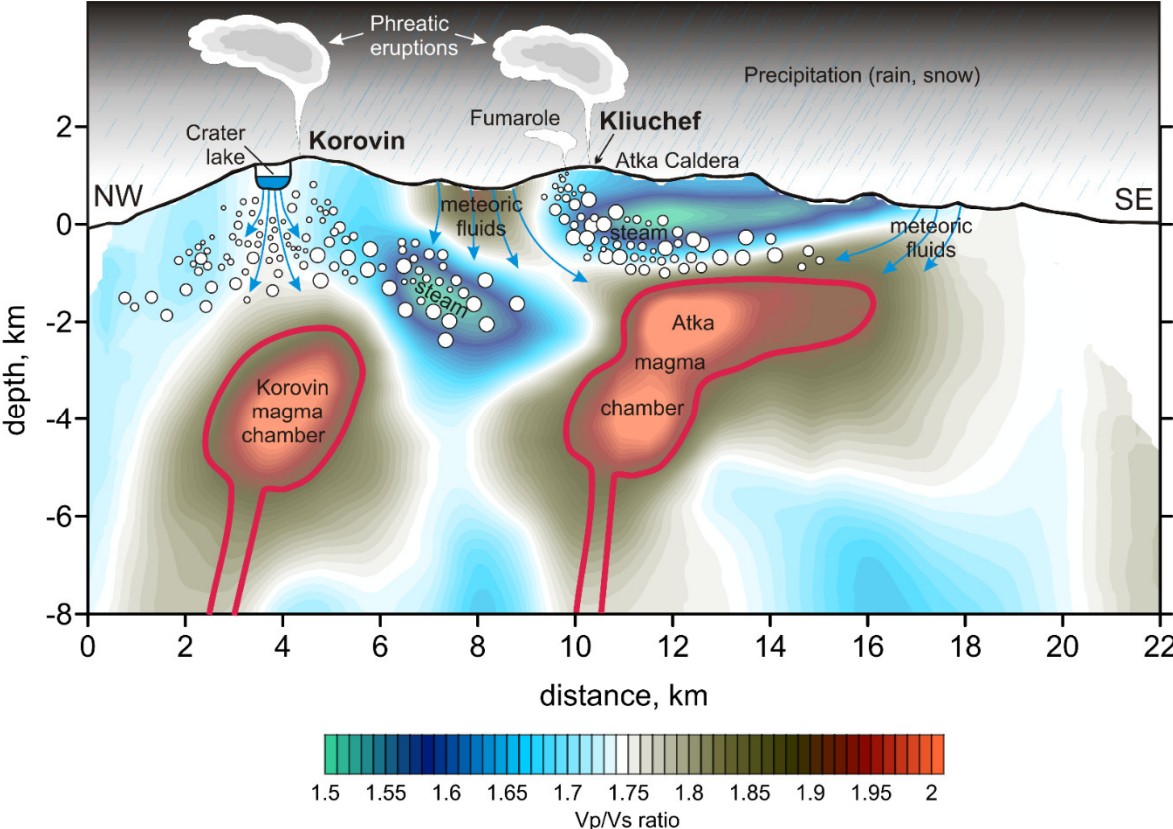

**Figure 7.** Conceptual representation of the magmatic and geothermal system beneath Atka Volcanic Complex (AVC), based on the distribution of the Vp/Vs in vertical section A1–B1 (Figure 6). Blue arrows indicate possible paths of downgoing flows of meteoric waters. White circles schematically depict the areas saturated with steam. Red contours highlight the active magma chambers beneath the Korovin and Kliuchef volcanoes.

Sarichev volcano, located at the eastern border of the island, has a regular conical edifice; however, there are not much information about its recent eruption activity. In our tomography model, it is located at the limit of the resolved area, and we cannot expect obtaining high-resolution structures there. Nevertheless, in section 2 in Figure 6, we observe a prominent anomaly of elevated Vp/Vs ratio centered with this volcano. The coexistence of high dVp and low dVs is an indicator showing that this might represent an intrusion of a magma with a high amount of dissolved fluids and/or with a high melting content. This might indicate that below Sarichev, there is still an active magma conduit that potentially can reset the eruption activity of this volcano.

As we see in Figure 1c, there was some activity on the flanks of the Atka shield volcano along the perimeter of the northeastern part of the island [15]. In addition to Korovin and Kliuchef, with clear indications of recent volcanic activity, there are at least six remnants of volcanic edifices and their eruption products indicated by M1 to M6, which were possibly active in the pre-caldera time (more than 0.3 Ma). In our tomography model, we see that beneath some of them some remnants of previously active conduits may still exist. For example, beneath Massif M2 (section 3 in Figure 6) we observe a vertical anomaly of moderately high Vp/Vs ratio, which may represent such an old conduit. Another Massif M5 (section 1) appears to be connected with the central conduit beneath the Atka Caldera, and was possibly fed by a lateral magma flow in the upper crust. The Massif M4 (section 4) seems to be associated with a shallow anomaly of a high Vp/Vs ratio, extending down to ~5 km depth. However, it appears to be implausible that any signatures of an active magma sources could kept at such shallow depths for hundreds of thousand years without any apparent feeding from below. Thus, it is more likely that these anomalies represent zones of saturation by meteoric or oceanic water. Beneath Massif M1, no apparent seismic anomaly is observed. This shows that, even if any conduit system existed there, it has now disappeared completely.

Based on the derived tomography model, in Figure 7, we present a conceptual image showing functioning of the active feeding system beneath AVC. This scheme is based on the distribution of the Vp/Vs ratio along the section A1–B1 in Figure 6 passing through the active Korovin and Kliuchef volcanoes, as well as the Atka Caldera. In this section, we see two large anomalies of high Vp/Vs ratio beneath Korovin and Atka Caldera, which are interpreted as two isolated shallow magma reservoirs beneath these volcanoes.

Previous experimental studies show that fractionation of AVC magmatic system started at ~26 km depth. It was probably a deepest level of magmatic reservoirs, where mostly basic magmas undergo limited crystal fractionation [37,42]. The major fractionation took place in shallower reservoirs located at 7–17 km depths [37], where magma evolved to more siliceous and high-alumina basalts, basaltic andesites, andesites and dacites. We suggest that this level of magmatic reservoirs was involved in complex magmatic history, described in [15]. The seismic anomalies described above are located at shallower depths within 1–5 km. They probably correspond to the third level of magmatic reservoirs proposed by [37], which were tapped by latest eruptions of Kliuchef and Korovin volcanoes.

We propose that in the shallow part of the model (down to 2–3 km below surface), the seismic anomalies are mostly controlled by migration and phase transformation of the meteoric fluids. Several spots of thermal activity have been observed on the western slopes of Kliuchef volcano and to the southwest of the Korovin volcano pediments. All of them are clear indication that there are hydrothermal systems confined to the Holocene volcanoes. The thermal activity areas lie at elevations from 225 to 1000 m a.s.l. The lowermost spots lie at the head of a wide valley 6 km to southeast of the Korovin volcano. They are associated with steam vents, thermal springs, mud pots and small explosive crater lakes. The mean temperature of the geothermal reservoir beneath this area is 169 °C, which is the lowest among other sources on Atka [17]. Thermal areas on the southwestern slope of the Kliuchef volcano lie at higher elevation (600–1000 m a.s.l.), and are related to reservoirs with higher mean temperatures 254–287 °C. Along with thermal springs and mud pots, these sites contain fumarolic vents, which are absent at the lowermost sites. The highest thermal site of the Kliuchef volcano is located below the Milky River glacier, which covers the summit of the volcano. According to Motyka et al [17], all hot sources in northeastern Atka contain sulfate-rich chlorine-poor waters, which may indicate that shallow meteoric water is heated by deeper boiling hot-water system associated with the magma sources, releasing steam, hydrogen sulfide and other gases. However, they do not present an accurate estimate for the ratio between the deep and shallow fluid sources. Thermal sites were also observed between Kliuchef and Korovin volcanoes [15], on the summit of Konia volcano and within southeastern crater of Korovon volcano.

A shallow anomaly of high Vp/Vs ratio between the Korovin and Kliuchev extends laterally to southwest from the pass between Kliuchef and Korovin volcanoes and is observed approximately to sea level depths. Such high Vp/Vs ratios are indicative for presence of liquids and at depths of above sea level may be associated with shallow groundwater. The same area of water accumulation might be associated with some increase of Vp/Vs in the right side of the profile. This part of the Aleutian Islands is characterized by rather humid climate bringing a large amount of the rain and snow water, which may penetrate underground. Another zone of locally elevated Vp/Vs ratio is observed beneath the summit of Korovin. On the top of one of the Korovin's summits, there is a large crater of 1 km size filled with a lake. It is known that, prior to large phreatic eruptions, this lake disappears [19]. The observed local anomaly might represent a permeable zone saturated with groundwater. Some of these waters may originate from seepage of the summit crater lake of the Korovin volcano. Notably, the thermal areas with lower mean temperatures of the reservoirs correspond to these high-Vp/Vs shallow anomalies. This agree well with the interpretations in [43].

The large "blue" anomalies of low Vp/Vs ratio, where very low Vp coexists with neutral or positive Vs, may represent areas of gas contamination. At the shallowmost section (Figure 5, section 0 km), most of the fumarolic activity seems to be associated with these low-Vp/Vs areas, or is located at the transition zones between the high- and low-Vp/Vs ratio. At depths down to 2–3 km (Figure 7), they mostly concentrate above and between the two magma reservoirs revealed as anomalies of high

Vp/Vs ratio below Korovin and Kliuchef. We propose that these areas of low Vp/Vs ratio mark steam-rich zones of the deep hydrothermal systems below these volcanoes.

Low-Vp/Vs anomaly under the Kliuchef volcano is located at higher elevations and almost outcrops to the surface right below the Milky River glacier. This can be explained by a low saturation of the ground with liquid water below the glacier at higher elevations. This steam reservoir is a probable source of gases for fumaroles and episodic phreatic eruptions of the Korovin and Kliuchef volcanoes. Boiling hydrothermal systems in volcanic edifices are usually isolated from ground waters by an impermeable clay layer, which appears due to hydrothermal alteration of primary volcanic rocks [36]. Assuming that this layer does not contain liquid water and steam, its presence should result in sharp increase in Vp/Vs ratio. This sharp gradient is clearly seen in Figure 6 sections A2–B2, A3–B3 and A4–B4 below the pass between Kliuchef and Korovin volcanoes and to southwest from Korovin volcano. Here, conductive heat transfer and steam seepage should heat groundwater, which give hot springs in the lowermost thermal areas. A clay envelope probably comes closer to the surface in the higher thermal areas of Kliuchef volcano and steam and gas seepage burst through the surface in the form of fumaroles, which are absent in the lowermost areas.

It can be seen in Figure 7 that the steam sources beneath Korovin and Kliuchev look as a unit system, which may deliver the gas to both volcanoes simultaneously. This may explain the facts of simultaneous phreatic eruptions of these volcanoes, such as one that was observed on 18 March 1987 [3].

## 5. Conclusions

Atka Island in the Aleutian is a hard-to-reach site, where the scientific investigations are hard to conduct. This fact makes it especially valuable the activity of the Alaskan Volcanological Observatory, which installed on this island eight permanent stations that operate in harsh climatic conditions from 2004. For the first time, we have used the data collected by this network to build a three-dimensional model of seismic P and S wave velocity and Vp/Vs ratio in the upper crust beneath a volcanic area in the Atka Island. Although eight stations is usually considered a low number for a high-quality tomographic inversion, in this study, by mean of performing a series of synthetic tests, we have proven that the available data geometry is sufficient to obtain appropriate resolution for revealing important information about the plumbing system beneath the Atka Island volcanoes.

The Atka Volcanic Complex (AVC) includes two active predominantly mafic volcanoes, Korovin and Kliuchef, which, in recent years, produced episodic phreatic eruptions and high thermal activity. Beneath each of these volcanoes we found anomalies of high Vp/Vs values reaching 2, which are interpreted as two isolated magma chambers in the upper crust. Beneath Korovin, the upper limit of the chamber is observed at the depth of 2 km below surface; beneath Kliuchef, it looks shallower—at 1.5 km depth.

At depths down to 2–3 km, the tomography model clearly reveals areas that can be interpreted as hydrothermally active. The model suggests that shallower level is probably saturated with liquid steam-heated waters, which feed relatively low temperature thermal springs located in the lowermost thermal areas to the west of Korovin and Kliuchef volcanoes. The deeper part of the hydrothermal system is most probably saturated with steam. It appears when meteoric waters penetrate deep below the surface and absorb the heat from shallow magmatic reservoirs of both volcanoes. The porous material saturated with steam is characterized by anomalies of very low Vp/Vs ratio reaching 1.5 that are observed in our model above the magma chambers. This steam returns to the surface from the shallowest system of Kliuchef volcano, and produces either episodic phreatic eruptions or the continuous fumarole activity.

We can conclude that the obtained tomography model has robustly identified the structures of both the magmatic sources and the meteoric fluid circulation in the uppermost crust. This result has allowed us to unravel a paradoxical problem related to independent geochemical properties of magmas outpoured from Korovin and Kliuchef, and, at the same time, an apparent interconnection between these two volcanoes when they produce synchronous phreatic eruptions. With our tomography model, we have proven that the Korovin and Kliuchef volcanoes are fed by magmas

from the isolated chambers, whereas the system of meteoric fluid circulation, which is responsible for phreatic eruptions, appears to be common in the entire volcanic system.

**Author Contributions:** I.K. wrote the major part of the article and took part in calculation of the tomography model; E.B. took the essential part in computiong the tomography model; S.Z.S. contributed in interpretation of the model and preparing the geological overview. All authors have read and agreed to the published version of the manuscript.

**Funding:** This study was supported by the RFBR grant # 18-55-52003.

**Acknowledgments:** We thank Jim Dixon from AVO who provided us the data for this study. The full directory of LOTOS with data corresponding to this study is available at www.ivan-art.com/science/atka.zip.

**Conflicts of Interest:** The authors declare no conflict of interest.

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
