# Peer review of "Magma Chambers and Meteoric Fluid Flows Beneath the Atka Volcanic Complex (Aleutian Islands) Inferred from Local Earthquake Tomography"

_geosciences, doi:10.3390/geosciences10060214_

Round 1

Reviewer 1 Report

Review of “Magma chambers and meteoric fluid flows beneath the Atka volcanic complex (Aleutian Islands) inferred from local earthquake tomography”

This is a well-written paper. The major contribution is to infer the magma chapter and meteoric fluid flows for Atka volcanic complex constrained using P-wave and S-wave velocity distributions. Technically, the manuscript uses seismic tomography to simultaneously estimate P- and S-wave velocities. The feasibility and resolution are verified using checkboard tests. For geological analysis, the authors give reasonable based on Vp, Vs and Vp/Vs ratio perturbations. Before publication, the author should consider the following comments and suggestions.

  • Page 6: give a brief introduction about which grid-search method you used for relocation. It is better to show the uncertainty for a couple of typical source locations in one iteration.
  • In figures 3-6: Why using different color bars to show Vp/Vs ratio and Vp and Vs perturbations? Are they more easily to see the differences?
  • Are the checkboard tests for 2D or 3D?
  • For the meteoric water, does this hypothesis have other evidence, like geochemical or geophysical data? Or, is it only yourself hypothesis? We should carefully check this hypothesis.

Overall, this paper is good. I suggest a minor revision. I have also attached a marked manuscript for further improving the language. You can use Adobe PDF reader to see the marked comments.

Author Response

Dear reviewer!

We are very grateful for careful reading our paper and valuable comments. For most of them, we have included corrections in the manuscript. Below, we present our responses to your questions and comments (indicated by REP and highlighted with bold).

I hope you will find our manuscript improved and suitable for further consideration.

Ivan Koulakov on behalf of the coauthors

Reviewer 1:

Review of “Magma chambers and meteoric fluid flows beneath the Atka volcanic complex (Aleutian Islands) inferred from local earthquake tomography”

This is a well-written paper. The major contribution is to infer the magma chapter and meteoric fluid flows for Atka volcanic complex constrained using P-wave and S-wave velocity distributions. Technically, the manuscript uses seismic tomography to simultaneously estimate P- and S-wave velocities. The feasibility and resolution are verified using checkboard tests. For geological analysis, the authors give reasonable based on Vp, Vs and Vp/Vs ratio perturbations. Before publication, the author should consider the following comments and suggestions.

Page 6: give a brief introduction about which grid-search method you used for relocation. It is better to show the uncertainty for a couple of typical source locations in one iteration.

REP: To address this comment, we have added a few sentences describing the principle of the grid-search method used in our case (L140-144).

In figures 3-6: Why using different color bars to show Vp/Vs ratio and Vp and Vs perturbations? Are they more easily to see the differences?

REP: According to this comment, we have changed the color scales for the Vp/Vs ratio in Figures 3 and 4 to make it consistent with Figures 5 and 6.

Are the checkboard tests for 2D or 3D?

REP: In both cases shown in Figures 3 and 4, the checkerboards are 2D. We find it more efficient to demonstrate the horizontal and vertical resolution in separate test series. In the text for the horizontal checkerboard, we write: “with depth, these anomalies remain unchanged” (L210). For the vertical test, we write: “Across the section, the anomalies have the width of 5 km and remain unchanged” (L222-223).

For the meteoric water, does this hypothesis have other evidence, like geochemical or geophysical data? Or, is it only yourself hypothesis? We should carefully check this hypothesis.

REP: The most comprehensive study of the Atka hot sources is contained in Motyka et al. (1981). They presume the coexistence of shallow and deep fluids, but do not provide any accurate estimates for their ratios. According to this comment, we have corrected a few sentences in the discussion as: “According to Motyka et al [17], all hot sources in northeastern Atka contain sulfate-rich chlorine-poor waters, which may indicate that shallow meteoric water is heated by deeper boiling hot-water system associated with the magma sources, releasing steam, hydrogen sulfide and other gases. However, they do not present an accurate estimate for the ratio between the deep and shallow fluid sources” (L364-368).

Overall, this paper is good. I suggest a minor revision. I have also attached a marked manuscript for further improving the language. You can use Adobe PDF reader to see the marked comments.

REP: Thank you very much for your corrections. They are all included into the updated manuscript.

Reviewer 2 Report

I attach my comments in a file.

Author Response

Dear reviewer!

We are very grateful for careful reading our paper and valuable comments. For most of them, we have included corrections in the manuscript. Below, we present our responses to your questions and comments (indicated by REP and highlighted with bold). 

I hope you will find our manuscript improved and suitable for further consideration.

Ivan Koulakov on behalf of the coauthors

Reviewer 2:

The article Magma chambers and meteoric fluid flows beneath the Atka volcanic complex (Aleutian Islands) inferred from local earthquake tomography is a very interesting seismic tomography study. The authors have used a not big dataset, however, the results are promising. The method has been widely used and it is well-known. The synthetic test show well results where the authors interpret the anomalies. The interpretation is in agreement with other geophysics studies done at the region. I have some comments, that are writing in the next lines.

Mistakes at text:

Line 56: the largest. (Add the before). Corrected.

Line 84: b. (instead b,). Corrected.

Line 130: 2. Data and algorithm Corrected.

Line 179: Sense. Corrected.

Line 193: 3. Tomography results. Corrected.

Line 220: The poorest. REP: Actually, we feel that in this case, in the phrase “the poorer resolution at this level…” the word “poorer” is more appropriate than “poorest”.

Figure 4: Please, indicate A1-B1 Figure 2. REP: We do not see where it should be corrected. All indications appear to be correct.

Line 252: 4. Discussion. Corrected.

Comments:

  1. Line 174: the nodes are placed in the regular grid with spacing of 1x1 km. However, in areas without data, the nodes are missing. In the vertical direction, the distance between nodes inversely depend on the ray density, but it cannot be smaller than a predefined value, 0.5 km in our case. Note that the predefined grid spacing is much smaller than the expected resolution of the model; every resolved anomaly is thought to be based on several nodes. In this sense, varying the parameters of the grid does not affect the resulting model. I understand that the grid spacing do not affect to the results. However I would like to know why do you chose 1x1km and 0.5km for vertical.

REP: This grid size appears to be optimal in our case. For the larger grid spacing, it would be compatible with the size of the resolved anomalies, and the model would be grid-dependent. The smaller grid spacing would lead to larger number of parameters, which would slow down the calculations. We have included the corresponding sentence in L159-161.

  1. I have been checking the program, where the authors have include the data and all the parameters to reproduce the tomography. I would like to thank this detail, it helps a lot to understand better this work. I have seen that in the initial model the include Vp/Vs constant with a value of 1.75. How do you chose this value?

REP: We use the constant Vp/Vs ratio in the starting model to avoid a predefined layering of this parameter. The value of 1.75 was defined after several trials of different reference models to achieve an appropriate balance between positive and negative anomalies in the resulting distributions of the dVp and dVs. We have added a paragraph with the description of the starting model definition (L179-188).

  1. The authors point that they have performed 5 iterations. How do you chose this value? What is the criteria?

REP: Five iterations were fixed as a compromise value to achieve sufficient quality of the results for reasonable calculation time. We have added a clarification on this issue in L177-178.

  1. At figure 5, for Vp and depth 7km, I have observed that through Korovin volcano appears a high velocity and low velocity anomaly, more or less, along B3-A3 section. How do the authors interpret this anomaly?

REF: We are sorry, but apparently we do not clearly understand the meaning of this comment. In the paper, we interpret the anomalies observed beneath Korovin at different depths in L238-263.

  1. My last comment is related to the seismicity distribution. I would like to know if the seismic tomography sections are plotted with the seismicity, if some patterns in the seismicity are observed, related with the anomalies. For example, in the authors interpret Atka magma chamber in two distinct parte: one from 2 to 4 km and another one from 4 to 6 km. Moreover, they related that with an intrusion of magma that took place from 2005 to 2007. Did you try to plot the seismicity for that period and observe the relation with the magma chamber

REP: We decided to present the seismicity in a separate Figure 2, because the seismicity clusters are rather dense and if we plot them over the resulting sections, they would strongly hide the velocity anomalies. Regarding to associating the seismicity with the upper and lower parts of the magma reservoir, as we can see in Figure 2, we cannot distinguish such separate clusters, therefore we do not stress our attention to this issue.